# Improvements in life expectancy mask rising trends in heat-related excess mortality attributable to climate change

Veronika Huber [1,2,3] ✉, Susanne Breitner-Busch[2,3], Hanna Feldbusch[3], Katja Frieler [4,5], Cheng He [3], Franziska Matthies-Wiesler [3], Matthias Mengel [4], Siqi Zhang[6], Annette Peters [2,3,7] & Alexandra Schneider[3]

Previous attribution studies of heat-related excess mortality have given limited attention to temporal trends in vulnerability and their non-climatic drivers. Here, we address this gap by combining counterfactual temperature data derived from multidecadal reanalysis series with time-varying warm-season temperature-mortality associations for the 15 most populous cities in Germany over 1993-2022. We find that declining vulnerability, associated with improvements in life expectancy, has led to decreasing trends in heat-related excess mortality in most cities despite summer warming. In contrast, if life expectancies had not improved, climate change would have induced increasing trends in the heat-related death burden. The growing anthropogenic fingerprint also emerges in the relative proportion of heat-related excess mortality attributable to climate change, which increased by 5.6% per decade (95% confidence interval: 2.6%, 8.6%), averaging 53.6 % (49.8%, 58.9%) across the study period. Our results underline the importance of accounting for evolving vulnerability when attributing human health outcomes to climate change.

Climate change over recent decades has already profoundly affected human health[1]. However, studies formally attributing observed changes in health outcomes to anthropogenic climate change are still relatively limited[2,3]. One of the most extensively studied climate-sensitive outcomes in the health attribution literature is heat-related excess mortality[4]. Past studies quantifying the burden of heat-related mortality attributable to climate change have mostly focused on specific extreme events[4–6] or considered multi-decadal averages[7,8]. To our knowledge, very few previous studies have attributed temporal trends in heat-related mortality to climate change, while accounting for changing temperature-mortality associations over time[9,10].

The evidence basis for past changes in population susceptibility to heat is comparatively large. The conclusion from recent reviews[11–13] and multi-country analyses[14,15] is that heat sensitivity has decreased over recent decades in most locations. Notable exceptions are reported increases in vulnerability to heat in Northern Europe[16] or the reversal in the trend of decreasing heat sensitivity observed in the Czech Republic[17,18], where unprecedented heatwaves occurred in the latter part of the study periods. Mixed evidence also exists for Germany, where studies investigating all-cause mortality have generally found decreasing population susceptibility to heat[19,20]. By contrast, increased heat sensitivity has been found for specific causes of

[1]Doñana Biological Station (EBD), Spanish National Research Council (CSIC), Sevilla, Spain. [2]Chair of Epidemiology, Institute for Medical Information Processing, Biometry, and Epidemiology, Faculty of Medicine, Ludwigs-Maximilian-Universität (LMU), Munich, Germany. [3]Institute of Epidemiology, Helmholtz Zentrum München – German Research Center for Environmental Health, Neuherberg, Germany. [4]Potsdam Institute for Climate Impact Research, Potsdam, Germany. [5]Institute of Environmental Science and Geography, University of Potsdam, Potsdam, Germany. [6]Department of Environmental Health Sciences, Yale School of Public Health, New Haven, CT, USA. [7]Munich Heart Alliance, German Center for Cardiovascular Health (DZHK e.V., partner-site Munich), Munich, Germany. ✉e-mail: veronika.huber@ebd.csic.es

cardiovascular and respiratory mortality[21], as well as for myocardial infarctions and strokes[22,23].

Since temperatures have increased nearly everywhere due to climate change, many locations have therefore experienced opposing trends of decreasing vulnerability versus increasing heat exposure over recent years. The current evidence suggests that declining relative risks (RR) have overcompensated for rising temperatures in a large number of these locations, with resulting negative trends in overall heat-related excess mortality[15,24]. However, given climate variability with differing patterns of heat extreme occurrence, trends are sensitive to the length and start of the chosen study period. Where the increase in heat extremes has been particularly strong in recent years, positive trends in heat-related excess mortality have also been observed[17,25].

Decreasing vulnerability has often been described as adaptation in the literature[26,27]. However, not all factors identified as possible causal agents can be considered part of an adaptive response to climate change. While there is evidence on the role of residential air conditioning[24,28], heat health action plans[29], and urban greening[30] in lowering population susceptibility to heat in some countries, non-adaptive factors such as socio-economic development, increased life expectancy and quality[31], and improved health-care services may also play an important role[32].

Our study is based on a dataset of daily all-cause mortality in the warm season (Jun-Sep) of 1993–2022 from the 15 most populous cities in Germany (> 0.5 Mio inhabitants). The dataset includes nearly 1.5 million deaths and represents 17.3% of the total German population. Series of daily mean temperatures covering 1950 to 2022 are obtained from the 2-m air temperature data of the ERA5-Land reanalysis by averaging the grid points that fall within the respective city boundaries. We first compute city-specific temperature-mortality associations in the warm season by 5-year subperiods (1993–1997 to 2017–2022), using distributed lag non-linear models[33] within quasi-Poisson time-series models. Subsequently, we pool these associations, while accounting for climatic, demographic, and socio-economic variables through multivariate meta-analytical techniques[34] (see "Methods").

We derive counterfactual temperature data, mimicking a world without anthropogenic climate change, from the available time-series of daily mean temperature by removing long-term trends in the average summer temperatures related to the observed rise in global mean surface temperatures (see "Methods"). In contrast to model-based approaches for obtaining counterfactual temperature data (e.g., used

in ref. 8), the chosen method reproduces observed temperature variability, allowing the derivation of attributable mortality specific to past extreme events or specific years[10,35]. Differences between factual (observed) and counterfactual temperatures, as well as the heat-related excess mortality derived from these temperatures[36], are considered attributable to anthropogenic climate change.

To assess the role of vulnerability changes, associated with improvements in life expectancy (LE), versus climate change (CC), we undertake all computational steps for computing attributable mortality twice. We use city-specific heat-mortality associations (i) as observed in each subperiod, and (ii) as predicted by keeping city-specific LE constant at the value of the first subperiod (1993–1997)[37]. We refer to these two setups as (i) with LE improvements, and (ii) without (w/o) LE improvements. Our study addresses an important research gap in the health impact attribution literature by disentangling climatic from non-climatic drivers of observed temporal trends in heat-related excess mortality.

## Results
### Climate-change attributable warming trends
The average warm-season temperature across cities over 1993–2022 was 17.4 °C (Supplementary Table 1) with an underlying linear warming trend of 0.68 °C (95% confidence interval: 0.33 °C, 1.02 °C) per decade (Fig. 1a). The counterfactual climate data suggests that, without climate change, temperatures would have only slightly increased over the study period, with a linear trend of 0.28 °C (95% CI: −0.07 °C, 0.62 °C) per decade (Fig. 1a). Across all years of the study period, we estimated 1.72 °C (range: 1.10 °C, 2.78 °C) of the average warm-season temperature to be attributable to climate change (Fig. 1a and Table 1). City-specific trends in observed temperatures (with CC) were significantly different from trends in counterfactual temperatures (w/o CC) (two-sided Wilcoxon rank test, $p < 0.001$) (Fig. 1b and Supplementary Fig. 1).

### Time-varying heat-mortality associations
Pooled (Fig. 2a) and city-specific (Supplementary Figs. 2 and 3) cumulative temperature-mortality associations in the warm season showed the usual pattern with increasing RR above the minimum mortality temperatures (MMT). The most parsimonious multivariate mixed-effect meta-regression model, from which we derived these associations (for details see "Methods"), included as meta-predictors besides average LE at birth (Fig. 2b), average annual temperatures, the number of potential heat alert days (as an indicator of heat extremes),

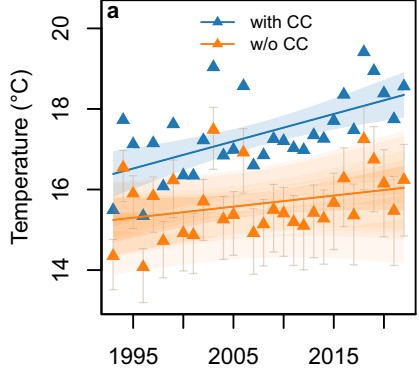
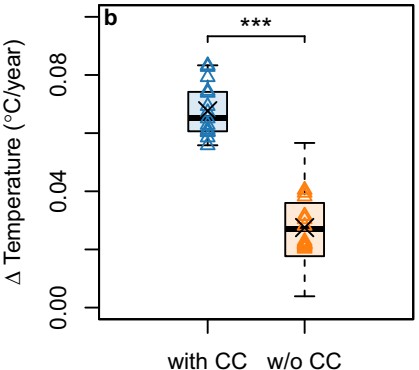

Fig. 1 | Contribution of climate change (CC) to trends in warm-season (Jun-Sep) mean temperatures in German cities (1993–2022). **a** Factual (blue) and counterfactual (orange) average annual warm-season temperatures across cities with linear regression lines (shading: 2 × standard errors); error bars (light shading) correspond to the uncertainty range (min, max) in counterfactual temperatures (trends), based on best estimates, lower and upper bounds of 95% confidence intervals for estimates of global mean surface temperature and local scaling

coefficients ($n = 9$ counterfactual temperature samples). **b** City-specific linear temporal trends; black crosses correspond to regression lines depicted in (**a**); boxplots show median (central line), upper and lower quartiles (box), and 1.5 × interquartile range (whiskers), taking into account counterfactual temperature uncertainty (see "Methods"); asterisks mark statistical significance level based on a two-sided Wilcoxon rank test ($p < 0.001$, combined $n = 15$ cities × (1 factual + 9 counterfactual temperature samples) = 150).

**Table 1 | Average climate-change attributable temperature increase, and average attributable annual heat-related death burden over 1993–2022, accounting for observed improvements in life expectancy**

| City name | Attributable warm-season temperature (°C) (min, max) | Sum of attributable heat excess deaths in 1993–2022 (95% eCI) | Annual attributable heat excess deaths ($AN_{CC}$) (95% eCI) | Attributable heat excess mortality fraction ($AF_{CC}$, %) (95% eCI) | Relative attributable proportion ($P_{CC}$, %) (95% eCI) |
|---|---|---|---|---|---|
| Berlin | 1.88 (1.22, 3.01) | 6762 (5302, 8328) | 225 (177, 278) | 2.01 (1.58, 2.48) | 57.4 (53.5, 62.3) |
| Bremen | 1.53 (0.96, 2.49) | 994 (689, 1275) | 33 (23, 43) | 1.56 (1.08, 20) | 52.7 (49.6, 57.0) |
| Cologne | 1.81 (1.18, 2.90) | 2484 (1938, 3011) | 83 (65, 100) | 2.60 (2.03, 3.15) | 52.8 (49.9, 57.2) |
| Dortmund | 1.70 (1.09, 2.74) | 1319 (998, 1617) | 44 (33, 54) | 2.04 (1.55, 2.51) | 53.4 (50.7, 57.2) |
| Dresden | 1.76 (1.13, 2.83) | 896 (509, 1264) | 30 (17, 42) | 1.68 (0.95, 2.36) | 56.4 (52.7, 62.0) |
| Duisburg | 1.69 (1.09, 2.73) | 1433 (1119, 1743) | 48 (37, 58) | 2.5 (1.95, 3.04) | 50.8 (47.0, 56.3) |
| Dusseldorf | 1.74 (1.13, 2.80) | 1467 (1112, 1828) | 49 (37, 61) | 2.24 (1.70, 2.80) | 52.2 (49.3, 56.0) |
| Essen | 1.69 (1.07, 2.73) | 1492 (1113, 1839) | 50 (37, 61) | 1.89 (1.41, 2.34) | 54.3 (52.1, 57.7) |
| Frankfurt | 1.84 (1.14, 3.02) | 1535 (1060, 2038) | 51 (35, 68) | 2.22 (1.53, 2.95) | 50.6 (45.6, 58.4) |
| Hamburg | 1.57 (0.97, 2.57) | 2370 (1704, 3078) | 79 (57, 103) | 1.29 (0.92, 1.67) | 52.6 (48.3, 58.4) |
| Hannover | 1.68 (1.06, 2.72) | 1664 (1132, 2165) | 55 (38, 72) | 1.43 (0.97, 1.86) | 54.2 (50.7, 59.3) |
| Leipzig | 1.81 (1.16, 2.92) | 1279 (838, 1733) | 43 (28, 58) | 1.96 (1.28, 2.65) | 54.8 (50.3, 61.8) |
| Munich | 1.67 (1.09, 2.65) | 2170 (1226, 3069) | 72 (41, 102) | 1.79 (1.01, 2.53) | 50.1 (42.5, 59.5) |
| Nuremberg | 1.67 (1.03, 2.75) | 1231 (943, 1511) | 41 (31, 50) | 2.12 (1.62, 2.60) | 50.2 (47.1, 55.3) |
| Stuttgart | 1.82 (1.18, 2.91) | 1411 (939, 1889) | 47 (31, 63) | 2.48 (1.65, 3.32) | 53.2 (47.8, 61.2) |
| All cities | 1.72 (1.10, 2.78) | 28506 (20758, 36363) | 950 (692, 1212) | 1.92 (1.40, 2.45) | 53.6 (49.8, 58.9) |

*eCI* empirical confidence interval.

and average population age (Supplementary Table 3 and Supplementary Fig. 4). Subperiod-specific, pooled RRs at the 99th percentile of daily warm-season temperatures showed a negative trend over the study period (Fig. 2c), suggesting a decreasing vulnerability of the German population to heat over time. Correspondingly, temporal trends in RRs derived from city-specific temperature-mortality associations were also negative in most cities (Supplementary Fig. 5). Model predictions based on LE fixed at the value of the first subperiod (1993–1997) suggested that heat-related RRs would have increased over the study period, had LE not risen at the same time (Fig. 2c). By contrast, predicted RRs continued to show a negative temporal trend when holding the other meta-predictors constant (Fig. 2c). Pooled (Fig. 2d) and city-specific temperature-mortality associations (Supplementary Figs. 3 and 5) indicated considerably higher RRs under the counterfactual assumption that LE had not improved since the first subperiod.

**Temporal trends in heat-related excess mortality**
Accounting for observed LE improvements, we found no significant difference (two-sided, Wilcoxon rank test, $p = 0.25$) between city-specific trends in heat-related mortality fractions (*AF*) based on factual (with CC) and counterfactual (w/o CC) temperatures (Fig. 3a, c). With the exception of one city (Supplementary Fig. 6), linear trend estimates of *AF*s were negative, even when factoring in the warming due to climate change (Fig. 3c and Table 2). On the other hand, when fixing LE at the level of the first subperiod (w/o LE improvements), trends in AFs were significantly different (two-sided Wilcoxon rank test, $p < 0.001$) in a world with and without climate change (Fig. 3b, d). In this case, under the assumption that population had been exposed to observed temperatures (with CC), trend estimates for heat-related *AF*s were exclusively positive over the study period (Fig. 3b, d, Table 2, and Supplementary Fig. 7). By contrast, no distinct temporal trend was observed (Fig. 3b, d and Table 2), when, in addition to fixed LE (w/o LE improvements), we assumed that no climate change (w/o CC) had taken place. The sign of these trend estimates was robust when considering absolute numbers of heat-related excess deaths and mortality rates (Supplementary Fig. 8 and Supplementary Table 4), and when changing model assumptions in the sensitivity analyses undertaken ("Methods", Supplementary Table 5).

**Climate-change attributable death burden**
On average, across the study period and across cities, we estimated that 1.92% (95% empirical CI: 1.40%, 2.45%) of warm-season all-cause mortality or 950 (95% eCI: 692, 1212) deaths per year were attributable to additional heat brought about by climate change (Table 1). In years, in which extreme heat waves occurred, the attributable death burden was considerably higher than the average (compare Fig. 1 and Fig. 4), with, e.g., in 2003, maximum values of 3.52% (95% eCI: 2.83%, 4.17%) of warm-season mortality, and 1746 (95% eCI: 1403, 2072) deaths summed across cities attributable to climate change (Supplementary Table 6). Over the entire 30-year study period, we estimated that in total 28 506 (95% eCI: 20 758, 36 363) heat-related deaths were attributable to climate change (Table 1). Our sensitivity analyses (see "Methods") showed that our estimates of the average heat-related death burden attributable to climate change did not depend on the specific model assumptions made (Supplementary Table 7).

Linear trend estimates for the absolute mortality fractions attributable to climate change ($AF_{CC}$) were close to zero, when changes in vulnerability associated with improvements in LE were accounted for (Fig. 4a, Table 2, and Supplementary Figs. 9 and 10). By contrast, if LE had not improved over time, one would have observed an increase in $AF_{CC}$ over time (Fig. 4a, Table 2, and Supplementary Figs. 9 and 10). The average relative proportion of heat-related excess mortality ($P_{CC}$) observed in 1993–2022 attributable to climate change was 53.6% (95% eCI: 49.8%, 58.9%) (Table 1), with an underlying linear trend of 5.6% (95% CI: 2.6%, 8.6%) per decade (Fig. 4b and Table 2). Assumptions on changing vulnerabilities, i.e., considering or not considering LE improvements, only slightly affected $P_{CC}$ and respective slope estimates (Fig. 4b, Table 2, and Supplementary Figs. 9 and 11).

## Discussion
This study documents a decreasing population vulnerability towards heat in the 15 most populous German cities from 1993 to 2022, associated with improvements in life expectancy. As a result, heat-related excess mortality has slightly declined over time, despite the climate-change induced warming trend. By contrast, under the counterfactual assumption of no improvements in life expectancy, climate change would have driven a rising trend in heat-related excess mortality. On average, we estimated that approximately 2% of warm-season all-cause

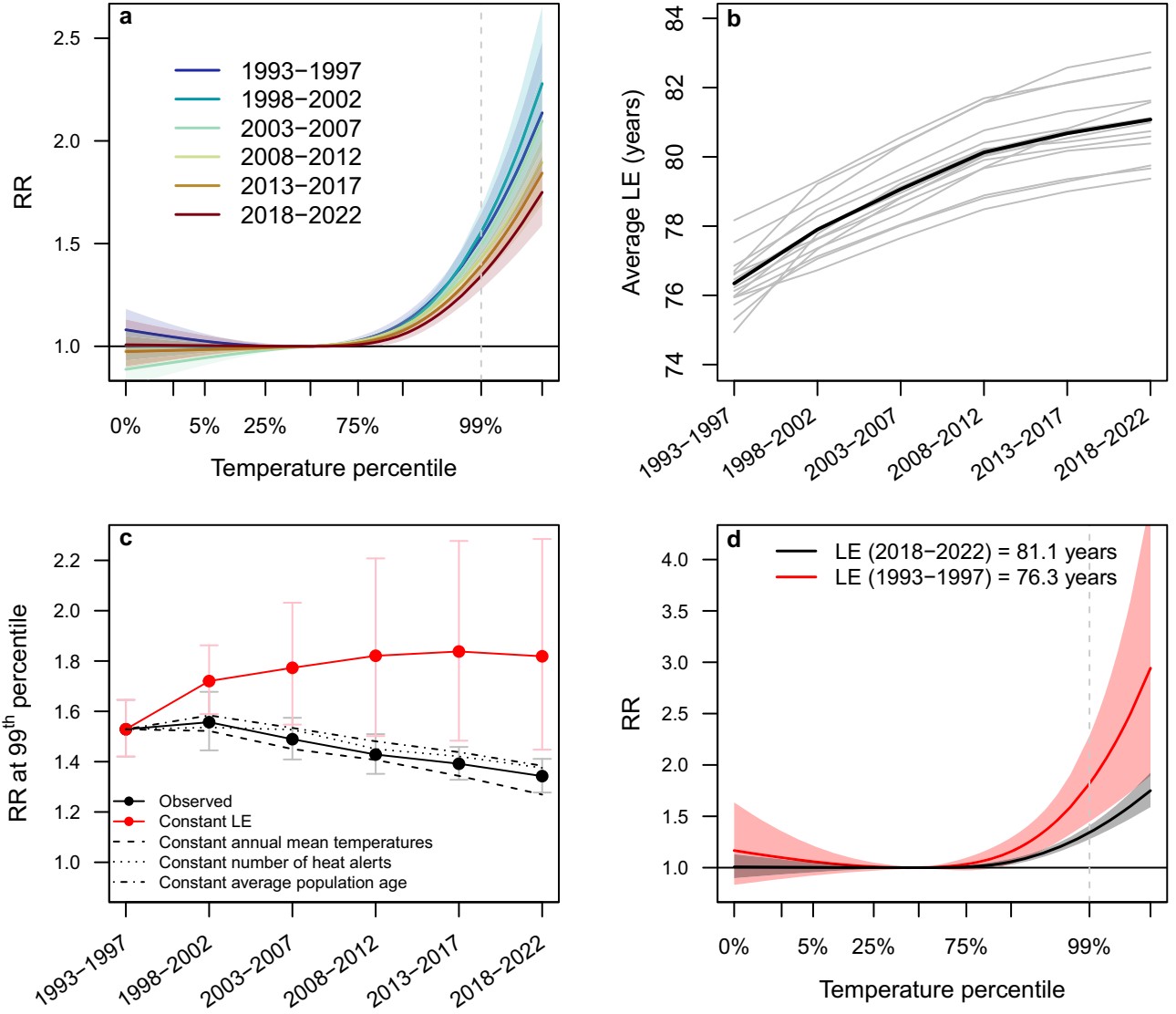

**Fig. 2 | Associations between improvements in life expectancy (LE) at birth and changing population vulnerability towards heat. a** Pooled cumulative temperature-mortality associations by subperiod derived from final meta-regression model (Supplementary Table 3). **b** Average LE by subperiod for each German city (grey) and averaged across cities (black). **c** Relative risk (RR) at the 99th percentile of temperatures (vertical dashed line in (**a**)); black dots: observed RR, red dots: predicted RR assuming no change in LE (fixed at 1993–1997 average), dashed and dotted black lines: predicted RR assuming no change in the other meta-predictors. **d** Pooled cumulative temperature-mortality association for 2018–2022, based on observed LE (black) and assuming LE had not improved since 1993–1997 (red). Error bands/bars (in **a**, **c**, **d**) correspond to 95% empirical confidence intervals (*n* = 1000 Monte Carlo samples).

mortality was attributable to heat exacerbated by climate change, with notably higher values in years with extreme summer heat waves. Overall, approximately 28 500 heat-related deaths could have been avoided over 1993–2022, if climate change had not increased warm-season temperatures in the studied cites. The relative proportion of heat-related excess mortality attributable to climate change in Germany's most populous cities has strongly risen over the last three decades, at a pace of around 6% per decade.

A previous study[9], investigating temperature-related mortality in England and Wales over 1976–2005, concluded that, only in the absence of adaptation, the human influence on climate would have been the main contributor to temporal increases in heat-related mortality. This aligns closely with our findings, considering that changes in the temperature-mortality associations referred to as adaptation in that study might at least partly be driven by non-climatic factors. Among the non-climatic factors considered in our study, we found that life expectancy is the best predictor of changing heat vulnerabilities

over time. Life expectancy gains may not only reflect an aging population but can be interpreted as a summary indicator of changes in population health[31,38]. Our results point to the latter interpretation: Improvements in general population health and medical care might underly the observed reduction in the relative risk of prematurely dying from heat exposure. This finding cautions against referring to the observed temporal trends in RR as adaptation, albeit a more refined analysis, including potential adaptive factors (such as heat health action plans, air conditioning, and urban greening) and further indicators of socio-economic development, would be warranted before drawing a firm conclusion.

Previously, 0.66% (95% eCI: −0.22%, 1.50%) of the warm-season all-cause mortality in German cities was reported to be attributable to climate-change-induced heat, corresponding to a relative proportion of 28.5%[8]. Our average estimates of the attributable mortality fractions (1.92 [95% eCI: 1.40%, 2.45%]) and relative proportion (53.6% [95% eCI: 49.8%, 58.9%]) were both higher. The divergence in estimates is likely

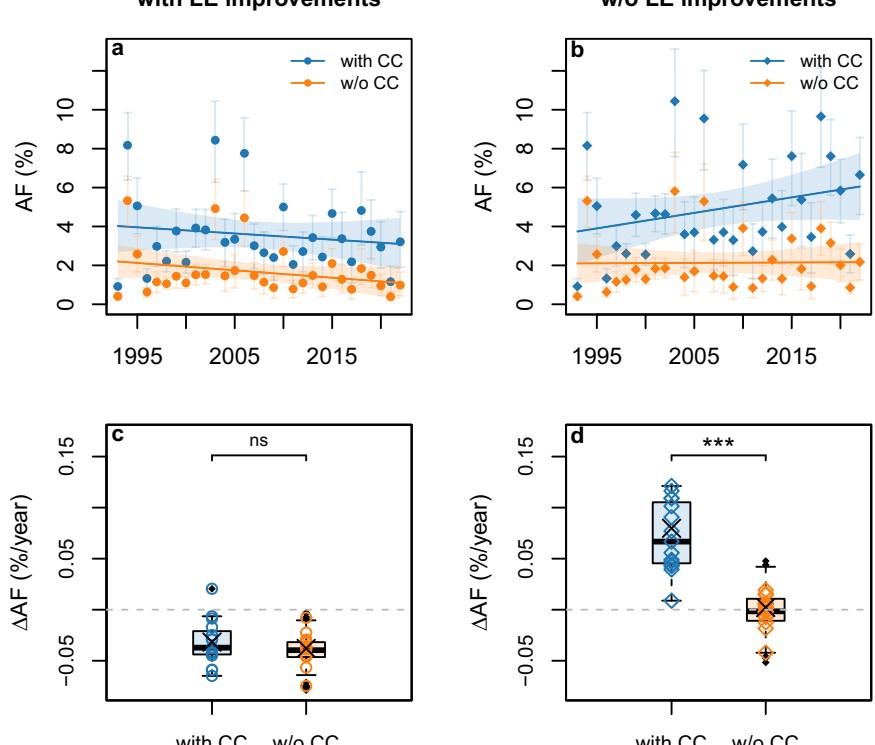

**Fig. 3 | Contribution of climate change (CC) to trends in heat-related excess mortality, with/without considering observed improvements in life expectancy (LE). a, b** City-average annual heat attributable fractions (AF) (as % of total warm-season mortality) for factual (blue) and best-estimate counterfactual (orange) temperatures; lines show linear regressions (with 2 × standard errors); error bars depict 95% empirical confidence intervals (n = 1000 Monte Carlo samples). **c, d** Corresponding city-specific linear temporal trends of heat AFs; symbols: central estimates (Supplementary Figs. 6 and 7); boxplots: median (central line), upper and lower quartiles (box), 1.5 × interquartile range (whiskers), outliers (black dots), taking into account counterfactual temperature uncertainty (see "Methods"); black crosses: regression lines in (**a, b**); statistical significance of differences in city-specific trends was assessed with a two-sided Wilcoxon rank test (***: p < 0.001; ns: p = 0.25, combined n = 15 cities × (1 factual + 9 counterfactual temperature samples) = 150).

**Table 2 | Linear slope estimates (with 95% confidence intervals) of city-average temporal trends in (climate-change attributable) heat-related excess mortality from 1993 to 2022 (see Figs. 3 and 4)**

| Temporal trend in heat-related excess mortality (% per decade) | | |
|---|---|---|
| | **With LE improvements** | **W/o LE improvements** |
| AF (with CC) | −0.32 (−1.13, 0.50) | 0.80 (−0.24, 1.84) |
| AF (w/o CC) | −0.38 (−0.90, 0.14) | 0.02 (−0.61, 0.65) |
| Temporal trend in heat-related excess mortality attributable to climate change (% per decade) | | |
| | With LE improvements | W/o LE improvements |
| $AF_{CC}$ | 0.06 (−0.26, 0.39) | 0.78 (0.33, 1.22) |
| $P_{CC}$ | 5.6 (2.6, 8.6) | 5.9 (2.7, 9.1) |

*LE* life expectancy, *CC* climate change, *AF* attributable fraction, $AF_{CC}$ attributable fraction due to climate change, $P_{CC}$ relative proportion due to climate change.

explained by the fact that the previous study[8] used model-derived counterfactual climate data in contrast to our purely observation-based approach. In addition, we considered time-varying temperature-mortality associations, whereas that study[8] used a single association representing the entire study period. Other recent attribution studies from Switzerland[5] and for 35 European countries[6] quantifying the heat-related death burden that could have been avoided in the absence of anthropogenic climate change, based on observational temperature counterfactuals, found proportions of 60% (in the summer of 2022

and 44% to 56% (in the period 2015–2022), respectively. These values are very similar to the average percentages of attributable mortality found here.

We would like to highlight three main strengths of our study. First, to the best of our knowledge, only very few studies[9,10] have previously investigated to what extent a change in vulnerability has dampened the contribution of climate change to the heat-related death burden. The terminology used in the health impact attribution literature is often unclear in this regard given that studies considering multi-decadal averages of excess mortality estimates[8] or cumulative heat-related excess deaths over multi-year periods[10] are categorized as "trend-to-trend" impact attribution[3], although temporal trends in the health outcome variable are generally not accounted for. Second, since the chosen method for deriving counterfactual temperatures reproduces observed climate variability[35], as one of its main advantages, we were able to single out specific extreme years occurring during the study period. Doing so, we found that, e.g., in the record-breaking summer of 2003, the heat-related death burden attributable to climate change in Germany's most populous cities was more than 1.5 times larger than the long-term average annual attributable burden. Third, we provide compelling evidence on the importance of accounting for changing vulnerabilities in assessments of past and future trends of heat-related excess mortality. There is a growing number of projection studies that consider possible future shifts in temperature-mortality associations[39–41]. We show that additional efforts to elucidate the adaptive and non-adaptive factors underlying observed changes in temperature-mortality associations are warranted to further improve the approaches taken so far.

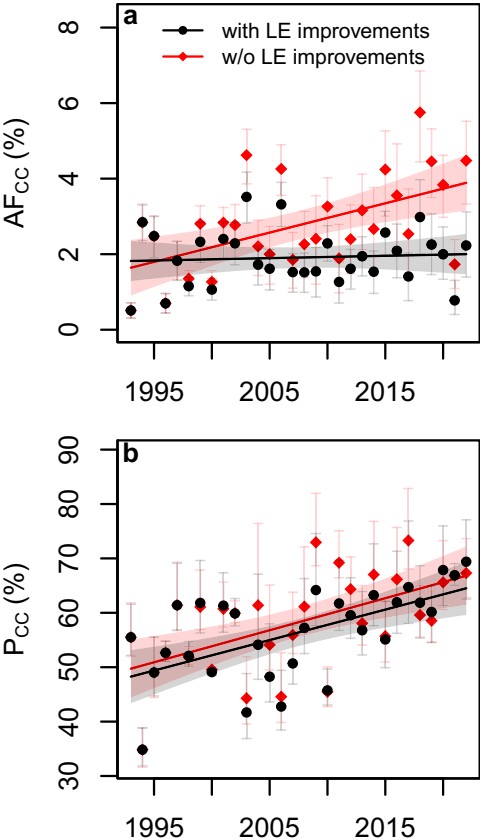

**Fig. 4 | Trends in heat-related excess mortality attributable to climate change (CC), with/without considering observed improvements in life expectancy (LE). a** Best-estimate, city-average attributable fractions (AF$_{CC}$) (% of total warm-season mortality) computed as the difference between factual and counterfactual estimates (see Fig. 3). **b** Best-estimate relative proportions (P$_{CC}$) of the observed (factual) heat-related excess mortality attributable to climate change; lines show linear regressions (with 2 × standard errors); error bars (in **a, b**) depict 95% empirical confidence intervals ($n = 1000$ Monte Carlo samples); see Supplementary Figs. 9–11 for corresponding city-specific trends.

A few limitations of our study also need to be acknowledged. First, our method for deriving counterfactual temperatures only accounts for regional temperature changes associated with global mean temperature rise. A recent climatological study[42] showed the importance of changes in circulation dynamics and aerosol forcing compared to background warming for explaining observed summer warming trends over western-central Europe. By ignoring aerosol-driven and dynamical changes when deriving temperature counterfactuals, we might misrepresent the more complex signal of anthropogenic climate change on regional warming in Germany. Second, the chosen observation-based approach to derive counterfactual temperature data did not allow us to explore the role of natural climate variability in driving observed temporal trends in heat-related excess mortality. Probabilistic assessments of temporal trends would require climate model-based approaches, such as the optimal fingerprinting technique applied in a previous study[9]. Third, due to missing age-specific daily mortality data, our analysis does not stratify by age and thus cannot account for demographic changes, which have been shown to influence temporal trends in temperature-related mortality[43,44]. The proportion of persons belonging to the highest age groups (65–74, 75–84, and 85+ years) in the German population has steadily increased over recent decades[20]. While the subperiod-average percentage of the population >65 years only appeared as a marginally significant

predictor (likelihood ratio test, $p = 0.081$) in the second-stage meta-regression (Supplementary Table 3), it is still conceivable that our estimates of the climate-change attributable death burden in Germany are biased by ignoring age-group-specific temperature-mortality associations. We expect this bias to be relatively low, given that population aging seems to affect cold-related mortality trends more strongly than heat-related mortality trends under climate change[44]. Future research, possibly taking an age-period-cohort approach[45], will need to further investigate observed shifts in heat vulnerability against the backdrop of medical progress, increasing life expectancies, and aging populations.

In conclusion, our study suggests that declining risks of heat-related mortality in the 15 most populous cities of Germany, associated with improvements in life expectancies, have so far overcompensated the increased heat exposure due to climate change. As a result, temporal trends in heat-related excess mortality were negative from 1993 to 2022. However, given the intense regional warming expected under current greenhouse gas emission trajectories, the trend may flip sign in the coming years. Strong efforts in adaptation, such as through heat-health action plans, urban greening, and the expansion of residential air conditioning, might lower the heat vulnerability of the population to keep up with the pace of increasing temperatures, which ultimately will depend on the stringency of mitigation undertaken.

## Methods
### Data sources
We accessed daily all-cause mortality data from the 15 most populous cities in Germany (>0.5 Mio inhabitants) in the period 1993–2022 through the Research Data Centre of the German Statistical Offices (summary statistics in Supplementary Table 1). Daily mean temperature data were obtained from ERA5-Land, averaging across the grid cells that fell within the city boundaries. We also computed annual average temperatures, temperature range, and average warm-season (Jun-Sep) temperatures from this data. In addition, we obtained annual socio-economic and demographic indicator data (originally derived from the INKAR database, Supplementary Table 2), and the number of potential heat alert days per year (as an indicator of extreme heat) from a previous analysis on German cities[46].

### Counterfactual temperature series
Counterfactual daily mean temperatures, mimicking a world without climate change, were computed by removing trends related to the observed global warming. Specifically, we first derived smoothed series of monthly anomalies of global mean surface temperatures (GMST) from HadCRUT5 (version 5.0.2.0)[47] by applying a singular spectrum analysis smoothing algorithm (with a window size of 120 months)[35,48]. We used 1850–1900 as a pre-industrial reference period, subtracting the corresponding mean from the original and smoothed GMST series (Supplementary Fig. 12). Subsequently, we computed linear regressions between the annual Jun-Sep average temperatures derived from ERA5-Land in each city and the annual averages of the smoothed GMST anomalies, including all years between 1950 and 2022 (Supplementary Fig. 13). Counterfactuals were then calculated by subtracting the product of the city-specific linear slope parameter and the smoothed monthly GMST anomalies from city-specific daily mean temperatures of each warm-season month over 1993–2022.

This method does not allow us to formally distinguish between natural and anthropogenic forcing underlying the observed rise in GMST. However, given that the contribution of natural forcings to the rise in GMST since the pre-industrial period is small, we assume that our approach approximates a counterfactual world without anthropogenic climate change. The approach chosen is very similar to the methods applied in recent non-probabilistic attribution studies of heat-related mortality[5,6]. The main difference is that these studies considered annual GMST anomalies attributable to anthropogenic

climate change to derive counterfactual daily temperature series only for one[5] or a very small number of years[6], while we used the smoothed monthly GMST series over the entire study period 1993–2022.

The main analysis is based on counterfactual temperature series derived from central estimates of HadCRUT5 and linear slope parameters. To enlarge the statistical power of tests on city-specific warming and mortality trends (see below), and to thereby strengthen the confidence that found differences were indeed attributable to climate change, we computed additional counterfactual temperature series, based on the upper and lower bounds of the 95% confidence intervals of GMST data (available from HadCRUT5) and of the city-specific linear slope estimates.

## Epidemiological models
We subdivided our data into six 5-year subperiods (1993–1997, 1998–2002, 2003–2007, 2008–2012, 2013–2017, 2018–2022) and restricted the analysis to Jun-Sep. We applied a two-stage design, first using quasi-Poisson regression models with distributed lag non-linear models (DLNMs)[33] to estimate time-varying warm-season temperature-mortality associations, and, subsequently, longitudinal multivariate mixed-effect meta-regression models to derive pooled model coefficients and identify factors related to the potential temporal trends in temperature-mortality associations[34]. Model specifications in the first stage were adopted from[8], using natural cubic splines with two inner knots for the exposure-response curve. These knots were fixed across all subperiods at the 50th and 90th percentiles of the overall distribution of daily mean temperatures during 1993–2022. We considered lags of up to 10 days and controlled for day of the week as well as seasonal and long-term trends with parameterizations as in ref. 8. Subsequently, for each city and subperiod, we computed reduced coefficients representing the risks cumulated over all lags and corresponding covariance matrices.

We pooled the reduced coefficients starting with an intercept-only model, including a random-effect per city (model0, Supplementary Table 3). To evaluate whether city-specific associations clustered by federal states or eastern/western Germany, we also tested corresponding nested random effects (model0.1, model0.2). We then added a natural spline function of the mid-point of each subperiod (year) (with one knot placed in the middle of the study period) to test for temporal trends (model0.3). We compared this model to an additional model, in which we also included random slopes for time (model0.4). We then added each climatic, demographic, and socio-economic meta-predictor (subperiod mean) in turn to model0.3, which had the lowest AIC (and BIC) from all simple models tested. We also used backward and forward selection to find the most parsimonious models with multiple meta-predictors, as proposed by ref. 34. We tested the significance of the included meta-predictors by performing likelihood ratio (LR) tests and computed $I^2$ statistics and Cochran's Q to test for residual heterogeneity.

From the final meta-regression model, we derived pooled model coefficients using the city-average of selected meta-predictors by subperiod. To assess the influence of the meta-predictors on temporal changes in temperature-mortality associations, we computed additional pooled associations fixing each meta-predictor in turn at the value of the first subperiod (1993–1997). As default, we used the average percentile temperature distribution across cities and subperiods to derive pooled temperature-mortality associations, which we centered on the 50th percentile. Using an absolute temperature scale, averaging daily mean temperatures across cities by date, resulted in similar pooled temperature-mortality associations (Supplementary Fig. 14). We computed RRs at the 99th percentile of the average warm-season temperature distributions to assess temporal trends in heat vulnerabilities.

Subsequently, we applied the final meta-regression model to derive (i) factual and (ii) counterfactual best linear unbiased predictors (BLUPs). Factual BLUPs were computed using standard approaches[34] and represented improved estimates of the observed city-specific temperature-mortality associations by subperiod. Counterfactual BLUPs were calculated as the sum of fixed-effect predictions, in which city-specific LE were held constant at the value of the first subperiod (1993–1997), and residuals from the factual BLUP computations[37]. We derived MMTs, as centering points of the city-specific curves, from both factual and counterfactual BLUPs by searching across the 25th to 99th percentiles of city-specific daily mean temperatures.

## Climate-change attributable mortality
We computed daily heat-attributable deaths during the warm season by city and subperiod, using forward moving averages of observed daily death counts across the lag period[36]. To assess the role of changing temperature exposure due to climate change versus changing life expectancy in these computations, we considered all combinations of subperiod-specific factual (with LE improvements) and counterfactual (w/o LE improvements) BLUPs, and factual (with CC) and counterfactual (w/o CC) daily mean temperatures, respectively.

For each setup, warm-season excess deaths attributable to heat (AN) were computed by summing across all days with temperatures > MMT. We also summed city-specific attributable numbers to arrive at aggregated estimates of excess mortality across cities, and divided by total warm-season mortality per year to compute heat-related mortality fractions (AF). The differences between warm-season estimates based on factual and counterfactual temperature data ($AN_{CC}$, $AF_{CC}$) were considered attributable to climate change. In addition to absolute differences in mortality fractions, we calculated attributable proportions ($P_{CC}$), dividing the absolute differences by the estimates based on factual temperature data. We computed 95% empirical confidence intervals (eCI), using Monte Carlo simulations with 1000 samples drawn from the assumed normal distribution of model coefficients[33].

## Assessment of temporal trends
To assess temporal trends in temperatures and attributable mortality estimates, we fitted linear regression lines. We also assessed non-linear alternatives, i.e., natural cubic splines with 2,3, and 4 degrees of freedom. However, these showed higher AICs than a simple linear model in most cases. We used two-sided Wilcoxon rank tests to evaluate the statistical significance of differences in city-specific linear trend estimates based on factual (with CC) versus counterfactual (w/o CC) temperatures, and factual (with LE improvements) versus counterfactual (w/o LE improvements) BLUPs, respectively.

## Sensitivity analyses
We conducted several sensitivity analyses. First, we used 5-year moving averages of annual GMST anomalies, rather than the monthly GMST data smoothed with singular spectrum analysis, to derive counterfactual temperature series. Second, we used daily mortality averaged by day of the year from the full study period instead of observed death counts in the computation of attributable mortalities. Third, we subdivided the data into five 6-year subperiods instead of using six 5-year subperiods. Fourth, we excluded the year 2003 from the analysis given its exceptional character in terms of extreme summer temperatures and observed heat-related excess mortality. All data analyses and plotting were performed within the R software environment (version 4.4.1) using the open-source packages dlnm (version 2.4.7) and mixmeta (version 1.2.0).

## Reporting summary
Further information on research design is available in the Nature Portfolio Reporting Summary linked to this article.

## Data availability

The city-specific daily mean temperature data was derived from gridded ERA5-Land reanalysis 2-m air temperature data available at the Copernicus Climate Data Store (https://cds.climate.copernicus.eu/datasets/reanalysis-era5-land?tab=download). Data of HADCRUT5 global mean surface temperature with monthly resolution can be downloaded from https://www.metoffice.gov.uk/hadobs/hadcrut5/. Demographic and socio-economic indicators for Germany can be accessed at https://www.inkar.de/. All-cause mortality data was obtained from the Research Data Centre (RDC) of the Statistical Offices of the German Federal States, which grant restricted data access to ensure statistical confidentiality and factual anonymity. The RDC provides data access upon conclusion of a user contract to institutions of higher education or other institutions tasked with independent scientific research, usually for a duration of three years (for details see https://www.forschungsdatenzentrum.de/en/terms-use).

## Code availability

R code used for data analysis and plotting is available at https://github.com/veronikahuber/TrendAttribution and permanently stored at https://doi.org/10.5281/zenodo.17303702. The code used to extract the city-specific daily mean temperature data from the ERA5-Land gridded dataset can be accessed at https://github.com/LAST-EBD/Consultas/tree/master/2025/Febrero/VH.

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

## Acknowledgements

We are thankful to Diego Garcia Diaz from Doñana Biological Station (EBD-CSIC) for setting up a Google Earth Engine routine and Python code for extracting city-specific temperature series from gridded ERA-5-Land data. V.H. is part of the Global Health Platform (PTI Salud Global) of the Spanish National Research Council (CSIC) and acknowledges funding from the European Union's Horizon 2020 research and innovation program (Marie Skłodowska-Curie Grant Agreement No.: 101032087), and the "Ramón y Cajal" fellowship program of the Spanish Ministry of Science and Innovation (RYC2022-036948-I).

## Author contributions

V.H., K.F., and M.M. were involved in conceptualization. V.H., S.B.-B., K.F., M.M., and A.S. designed the methodology. V.H. conducted the formal analysis. V.H., H.F., M.M., A.P., and A.S. were involved in resources and data curation. V.H. undertook visualization. V.H. wrote the draft manuscript. V.H., S. B.-B., H.F., K.F., C.H., F.M.-W., M.M., A.P., S.Z., and A.S. reviewed the manuscript. V.H., A.P., and A.S. acquired funding. A.P. and A.S. supervised the project.

## Competing interests

The authors declare no competing interests.
