## [Transparent Peer Review file · Nature Communications]

Improvements in life expectancy mask rising trends in heat-related excess mortality attributable to climate change

Corresponding Author: Dr Veronika Huber

Version 0:

Reviewer comments:

Reviewer #1

(Remarks to the Author)

The study provides compelling evidence of decreasing heat vulnerability in German cities from 1993 to 2020, largely attributed to improvements in life expectancy and population health. However, climate change remains a significant driver of heat-related mortality, with approximately 18,000 deaths over the study period linked to climate-induced warming. These findings highlight the complex interaction between adaptation measures and the continued effects of climate change, offering a detailed analysis of mortality trends influenced by anthropogenic factors.

By integrating observational data and counterfactual scenarios, the study addresses key gaps in attribution research, particularly in analyzing temporal trends in heat-related mortality and adaptation. The methodology is robust, utilizing a dataset of over 1.2 million deaths across 13 cities, with sensitivity analyses validating the findings. However, counterfactual scenarios focus only on regional temperature changes linked to global temperature rise, potentially underrepresenting the full impact of anthropogenic climate change.

Recommendations:

1. The distinction between "adaptation" and "w/o adaptation" periods is defined in the paper. While it is important to consider when data availability begins, a key question arises regarding the certainty with which we can claim that there was no adaptation during the initial period (1993–1996) used in the "w/o adaptation" setup. How can the potential influence of early adaptation efforts be accounted for or disentangled from this baseline assumption? Addressing this could strengthen the clarity and robustness of the conclusions.
2. A more detailed analysis of regional variations (e.g., differences between eastern and western German cities) and temporal trends could provide additional insights into localized adaptation capacities and vulnerabilities.
3. Citing more recent studies on adaptation and its influence on heat-related and climate-related mortality (HRM and CRM) would situate the findings more comprehensively within the broader research framework.

Suggestions for future research:

Future studies could explore more adaptation measures, such as investigating the role of air conditioning prevalence or access to green spaces, which may significantly influence heat vulnerability and adaptation.

Questions for the authors:

Have you observed any significant differences in heat-related mortality between sexes? This could provide additional insights into potential disparities in vulnerability and inform targeted adaptation measures.

The methodological section is well-documented, with a wealth of supplementary materials and tables to support transparency. I recommend the article for acceptance with minor revisions to clarify and enhance certain aspects.

(Remarks on code availability)

Reviewer #2

(Remarks to the Author)

(Remarks on code availability)

Reviewer #3

(Remarks to the Author)

Thank you for attribution analyses that will make an interesting contribution to the literature.

The abstract and discussion state that decreased vulnerability was because of increases in life expectancy, but the manuscript did not clearly describe how the analyses differentiated between life expectancy and other factors driving vulnerability. How much did life expectancy increase across the study period? Was it uniform across the age range? It was not clear why analyses including life expectancy were labeled as adaptation -- changing life expectancy is not an adaptation to climate change.

The manuscript refers to adaptation, changing vulnerability, and changing life expectancy. These are the same?

It would be more appropriate to use mortality rates than absolute number of cases, which depends on population size.

The manuscript states the analyses were not stratified by age, but then immediately reports analyses for adults >65 years.

The decadal analyses showing changing proportional of attributable mortality needs to be clearer.

How frequent were extreme heat events on an annual basis?

The key analyses should be replicated removing the year 2003, to determine the extent to which the heatwaves that year affected the results.

Why knots at the 50th and 90th %iles?

Line 37: temperatures have not ubiquitously increased.

Line 224: what is the small number?

(Remarks on code availability)

Reviewer #4

(Remarks to the Author)

Decreasing vulnerability masks rising trends in heat-related excess mortality attributable to climate change

The manuscript deals with heat related mortality risks in temperate areas. It seeks to separate the trends of decreasing heat-vulnerability and attributed and observed impacts from climate change. It finds vulnerability has decreased over the last 30 years in 8 cities of Germany, but mortality to heat has still increased due to climate change. This is an important finding. Many studies has documented the reduced vulnerability to heat over time before, but fewer has tried to explain it. The study is very well written and overall well presented, but this is a complicated matter and at this population aggregated level many things could probably modify and co-drive the patterns observed. Not all of them has likely been addressed here.

Major comments:

- The study is using increased life expectancy to explain the attenuation of heat related mortality. However, the attribution to life expectancy is difficult. Likely many factors could have explained the change. What has been investigated in terms of sensitivity to this choice? I could not find much discussion on this? How come life expectancy is coming out important whilst older age is likely the biggest risk factor to heat-related mortality? Do the results hold even if age and sex are stratified on? Presumably the mean age of death is different in the different study periods and the underlying causes of death.
- Tables and Figures are using with and without adaptation in both rows and columns. It makes it difficult to grasp at first. It should be better explained and maybe one can focus on presenting some of the results in a simplified version.
- Some tables include p-values and most of them appear non-significant. What do these correspond to? Currently it appears to show non-significant differences while the authors present the findings as if they do observe a difference and trend.
- It is understandable to want to keep the climate variability in the counterfactual analyses, but I would like to have more data and graphics on the attribution of pre-industrial to current time.

(Remarks on code availability)

Reviewer #5

(Remarks to the Author)

(Remarks on code availability)

Reviewer #6

(Remarks to the Author)

Thank you for the opportunity to review this interesting article. The manuscript is well written, researches a relevant topic and uses a novel approach to provide a better understanding on how climate change and other social changes shaped temperature related mortality in the past 30 years in Germany.

Here I provide some comments to the authors:

- The title could be improved to better convey the message of the paper. For instance, the authors could consider something on the lines of "Improvements in life expectancy compensate for increasing risks in heat-related excess mortality attributable to climate change" ;
- In the abstract, the authors could stress more the implications of their findings that currently are not evident. Similarly, the authors could improve discussion of the implications in the discussion section as well. For instance, the authors could enlarge the discussion on how their findings could inform studies that project future trends in temperature related mortality.
- The authors use data from KNMI for temperature. However, the authors could provide further details on why this dataset is chosen compared to other gridded datasets such as ERA-5 or E-OBS provided by the Copernicus Data Store that might provide better coverage. For instance, the KNMI data has a large number of missing data on temperature.
- The authors used a different approach compared with previous studies in climate attribution. Additional information, should be provided in the methods section and/or in the appendix on their data management procedures and on the advantages and disadvantages compared with previous approaches. Currently, it is not very easy to understand the steps they took and how sensitive these are to potential alternative choices.

(Remarks on code availability)

A Readme file is not currently present and could be added by the authors. The code has relevant title but could benefit from additional comments in some parts to better guide use. Also, as the data is not provided (at least in the files I reviewed) the authors could provide additional guidance in the code on where to access this.

Version 1:

Reviewer comments:

Reviewer #1

(Remarks to the Author)

The authors have thoroughly and constructively addressed the reviewers' comments, significantly improving the manuscript. The revised version presents clearer methodological distinctions and expanded results. Including new figures/improved labeling enhances the transparency and interpretability of the study. The authors also provided answers to the questions and issues raised during the review process. I consider the revised manuscript suitable for publication and recommend its acceptance.

(Remarks on code availability)

Reviewer #2

(Remarks to the Author)

(Remarks on code availability)

Reviewer #3

(Remarks to the Author)

Thank you for comprehensively addressing my comments.

(Remarks on code availability)

Point-by-point responses to reviewer comments on NCOMMS-24-65129

Please note that all line references given in our responses correspond to the clean versions of the manuscript and supplementary material.

Reviewer #1 (Remarks to the Author):

The study provides compelling evidence of decreasing heat vulnerability in German cities from 1993 to 2020, largely attributed to improvements in life expectancy and population health. However, climate change remains a significant driver of heat-related mortality, with approximately 18,000 deaths over the study period linked to climate-induced warming. These findings highlight the complex interaction between adaptation measures and the continued effects of climate change, offering a detailed analysis of mortality trends influenced by anthropogenic factors.

By integrating observational data and counterfactual scenarios, the study addresses key gaps in attribution research, particularly in analyzing temporal trends in heat-related mortality and adaptation. The methodology is robust, utilizing a dataset of over 1.2 million deaths across 13 cities, with sensitivity analyses validating the findings. However, counterfactual scenarios focus only on regional temperature changes linked to global temperature rise, potentially underrepresenting the full impact of anthropogenic climate change.

R1.1: We thank the reviewer for the positive overall evaluation of our study. We agree regarding the caveat of potentially underrepresenting the full impact of anthropogenic climate change, as discussed in ll. 198-204 of the revised manuscript. It is important to point out, however, that the main strength of our analysis is the disentangling of climatic and non-climatic factors underlying the temporal mortality trends. Given the demonstrated importance of the non-climatic drivers, we believe that our results are robust even if our estimation of the anthropogenic contribution of past warming trends was – due to the simple approach for deriving temperature counterfactuals – under- (or over-) estimated. As a new addition in the revised manuscript, we also consider uncertainty in counterfactual temperatures (ll. 258-263, Fig. 1), better reflecting the uncertainty regarding attributable warming.

Recommendations:

1. The distinction between "adaptation" and "w/o adaptation" periods is defined in the paper. While it is important to consider when data availability begins, a key question arises regarding the certainty with which we can claim that there was no adaptation during the initial period (1993–1996) used in the "w/o adaptation" setup. How can the potential influence of early adaptation efforts be accounted for or disentangled from this baseline assumption? Addressing this could strengthen the clarity and robustness of the conclusions.

R1.2: In principle we agree with this thoughtful comment. However, in the revised manuscript, we have redefined the 'non-climatic scenarios' investigated (see ll. 69-76 of revised manuscript). Instead of considering a "w/o adaptation" setup, we now take the much more explicit counterfactual assumption of no improvement in life expectancy since the first subperiod (1993-1997). Therefore, the concern raised here about early adaptation efforts is no longer relevant. See also response R3.1.

2. A more detailed analysis of regional variations (e.g., differences between eastern and western German cities) and temporal trends could provide additional insights into localized adaptation capacities and vulnerabilities.

R1.3: In the revised supplementary material (Model 0.1 and 0.2 in Supplementary Table 3), we included the results for additional meta-regression specifications, in which we included nested random effect for cities clustered by eastern/western Germany or federal states. Comparing AICs suggested a slightly improved fit for the model accounting for the clustering in federal states. Yet, based on this analysis no indication for systematic differences in the associations between eastern and western cities were found.

3. Citing more recent studies on adaptation and its influence on heat-related and climate-related mortality (HRM and CRM) would situate the findings more comprehensively within the broader research framework.

R1.4: In the revised manuscript, we have included four studies published in 2024 (references: 17, 26, 27, 31) to reflect more recent advancements and approaches to including adaptation in studies of heat-related mortality.

Suggestions for future research:

Future studies could explore more adaptation measures, such as investigating the role of air conditioning prevalence or access to green spaces, which may significantly influence heat vulnerability and adaptation.

R1.5: We fully agree with this suggestion, as reflected in ll. 166-168 of the manuscript, stating that “a more refined analysis, including potential adaptive factors (such as heat health action plans, air conditioning, and urban greening) and further indicators of socio-economic development, would be warranted [...]”

Questions for the authors:

Have you observed any significant differences in heat-related mortality between sexes? This could provide additional insights into potential disparities in vulnerability and inform targeted adaptation measures.

R1.6: We agree that stratifying the analysis by sex would be a valuable step for improving our understanding of potential differences in vulnerability between women and men. However, unfortunately, we only had access to overall death count data, not allowing for stratification by sex, nor by age. See also response R4.1.

The methodological section is well-documented, with a wealth of supplementary materials and tables to support transparency. I recommend the article for acceptance with minor revisions to clarify and enhance certain aspects.

R1.7: The revised supplementary material includes additional tables and figures (e.g., regarding the derivation of counterfactual temperature data, Supplementary Figs. 10 and 11), which should further increase transparency and documentation of methods.

Reviewer #2 (Remarks to the Author):

Reviewer #3 (Remarks to the Author):

Thank you for attribution analyses that will make an interesting contribution to the literature.

The abstract and discussion state that decreased vulnerability was because of increases in life expectancy, but the manuscript did not clearly describe how the analyses differentiated between life expectancy and other factors driving vulnerability. How much did life expectancy increase across the study period? Was it uniform across the age range? It was not clear why analyses including life expectancy were labeled as adaptation -- changing life expectancy is not an adaptation to climate change.

R3.1: We thank the reviewer for this important comment. Instead of generically considering changes in vulnerability as in the previous version of the manuscript, the revised analysis allows us for explicitly exploring the role of improving life expectancies over the study period (ll. 69-76 and ll. 298-305). We changed Fig. 2 to show the observed increases in life expectancy: On average, across all 15 cities, life expectancy increased by almost 5 years between the first and the last subperiod (Fig. 2b). In addition, in Fig. 2d, we illustrate the effect of improved life expectancy on the temperature-mortality association. Overall, we no longer label the scenarios considered 'with/without adaptation', but we now refer to the new analyses as 'with/without LE improvements' (see Figs. 3 and 4).

The manuscript refers to adaptation, changing vulnerability, and changing life expectancy. These are the same?

R3.2: We fully agree that a weakness of the previous analysis was to not distinguish well enough between these three concepts. In ll. 46-51, we explain why adaptation is not the same as changing vulnerability. As explained above, the new analysis then allows us for separating the role of improving life expectancy contributing to the overall changes in vulnerability observed. We believe that with the new setup there is less risk of confusion between the named concepts.

It would be more appropriate to use mortality rates than absolute number of cases, which depends on population size.

R3.3: We present the main results for attributable mortality fractions (AF), which are corrected for changes in the mortality baselines/population size (Figs. 3 and 4). Nevertheless, we have added two extra figures in the supplementary material (Supplementary Fig. 6c, d), showing mortality rates (per 100 000 population) to make sure that there is no bias from changing population sizes when presenting results on total attributable death counts (AN). Results for the different attributable mortality measures (fractions, numbers, rates) are very similar, because, considering that attributable numbers and population size differ by several orders of magnitude, observed changes in the population size are comparatively small.

The manuscript states the analyses were not stratified by age, but then immediately reports analyses for adults >65 years.

R3.4: We could not stratify the analyses by age, due to missing age-specific daily mortality data. The only data on demographic changes we had access to was annual data on the % population > 65 years,

which we included in the second-stage meta-analysis (cf. Supplementary Table 3). We slightly rephrased ll. 208-219 to explain better which type of age-specific data was available and to avoid confusion about why we did not stratify the analyses by age.

The decadal analyses showing changing proportional of attributable mortality needs to be clearer.

R3.5: We agree that the previous presentation of results on attributable mortality trends might have been confusing. In the revised manuscript, we have simplified Fig. 4 (moving part of it to Supplementary Fig. 7). See also response R4.3.

How frequent were extreme heat events on an annual basis?

R3.6 The revised supplementary material includes information on the annual number of extreme heat events, represented by the number of (potential) heat alerts, in Supplementary Fig. 3. On average, the cities experienced between 5 and 10 extreme heat days per year.

The key analyses should be replicated removing the year 2003, to determine the extent to which the heatwaves that year affected the results.

R3.7: As suggested by the reviewer, we have implemented a sensitivity analysis, in which we have removed the year 2003 (shortening the third subperiod to 2004-2007). The results are reported in Supplementary Tables 5 and 7. The only noticeable difference to the default setup was that the positive linear trend estimate under the assumption of no improvements in life expectancy ('w/o LE improvements) and warming due to climate change ('with CC') was greater than in the default case (and the 95% CI no longer included zero). Hence, removing the year 2003 from our analysis revealed an even stronger imprint of climate change on trends in heat-related excess mortality. Yet, overall, we found that our results were not substantially altered by excluding the year 2003.

Why knots at the 50th and 90th %iles?

R3.8: We adopted the parameterisations of the first-stage models from Vicedo-Cabrera et al. 2021 (reference 8), because we wanted to make our results as comparable as possible to the estimates of that study, which, to our knowledge, constitute the only previous estimates of the heat-related excess mortality attributable to climate change in Germany. It is important to note that setting the exposure knots at the 50th and 90th percentiles of the temperature distribution is a common choice when modelling temperature-mortality associations in the warm season (e.g., also taken in Gasparrini et al. 2015, reference 13).

Line 37: temperatures have not ubiquitously increased.

R3.9: We agree. There are a few exceptions. The sentence now reads "Since temperatures have increased nearly everywhere..." (l. 38)

Line 224: what is the small number?

R3.10: We included a new table in the supplementary material (Supplementary Table 2), in which we report in detail the existing data gaps with regard to the socio-economic and demographic data used. We deleted the imprecise expression 'small number' from the revised manuscript (ll. 234-236).

Reviewer #4 (Remarks to the Author):

Decreasing vulnerability masks rising trends in heat-related excess mortality attributable to climate change

The manuscript deals with heat related mortality risks in temperate areas. It seeks to separate the trends of decreasing heat-vulnerability and attributed and observed impacts from climate change. It finds vulnerability has decreased over the last 30 years in 8 cities of Germany, but mortality to heat has still increased due to climate change. This is an important finding. Many studies has documented the reduced vulnerability to heat over time before, but fewer has tried to explain it. The study is very well written and overall well presented, but this is a complicated matter and at this population aggregated level many things could probably modify and co-drive the patterns observed. Not all of them has likely been addressed here.

R4.1: We agree with the reviewer that disentangling the drivers of observed vulnerability and mortality trends is complex. And some mechanisms might indeed not be revealed at the population aggregated level. Yet, unfortunately, we don't have access to disaggregated data allowing e.g., for stratified analyses by sex and age. We think, however, that our analyses could incentivize future more detailed studies that could then verify and possibly elucidate patterns underlying the associations found in our study. The innovative contribution of our study, even without including stratified analyses by age and sex, is that we show the importance of separating climatic from non-climatic factors in the attribution of climate change impacts on human health.

Major comments:

- The study is using increased life expectancy to explain the attenuation of heat related mortality. However, the attribution to life expectancy is difficult. Likely many factors could have explained the change. What has been investigated in terms of sensitivity to this choice? I could not find much discussion on this? How come life expectancy is coming out important whilst older age is likely the biggest risk factor to heat-related mortality? Do the results hold even if age and sex are stratified on? Presumably the mean age of death is different in the different study periods and the underlying causes of death.

R4.2: In the revised manuscript, we have tried to better explain our focus on life expectancy, which we found to be strongly associated with observed changes in vulnerability to heat over time (II. 90-105). In fact, we tested 13 different climatic, socio-economic, and demographic meta-predictors (Supplementary Table 3). The best model, based on step-forward selection, included besides life expectancy, average annual temperatures, an indicator of heat extremes, and indeed also the average population age (mean age of death was not available). Yet, predictions of RRs, holding each of these meta-predictors constant in turn, suggested that life expectancy is the most prominent factor underlying the observed decreases in vulnerability towards heat over time (Fig. 2c).

Given the strong collinearity of factors potentially explaining the changes in heat vulnerability over time, we cannot, of course, exclude that drivers other than life expectancy gains have been equally important in driving the observed trends in heat-related mortality risk. As suggested by the reviewer an age-stratified analysis would be key to disentangle the roles of older age intricately linked to higher survival probabilities/life expectancy. In the revised manuscript, we have suggested to use an age-period-cohort approach, as recently suggested by Yuan et al. 2025 (reference 46), to shed more light on these complex relationships (II. 217-219).

- Tables and Figures are using with and without adaptation in both rows and columns. It makes it difficult to grasp at first. It should be better explained and maybe one can focus on presenting some of the results in a simplified version.

R4.3: In the revised manuscript we have simplified Fig. 4, moving part of the previous figure in the appendix (corresponding to the new Supplementary Fig. 7). Doing so allowed us to present results with and without accounting for life expectancy gains (the equivalent to the former comparison of with and without adaptation) consistently by columns (cf. Table 2 and Fig. 3). We hope that with this simplification the results are easier to understand.

- Some tables include p-values and most of them appear non-significant. What do these correspond to? Currently it appears to show non-significant differences while the authors present the findings as if they do observe a difference and trend.

R4.4: We agree that the previous version of the manuscript suffered from some incongruence regarding the presentation of significant versus non-significant temporal trend estimates. Given the relatively small number of data points ($n = \text{number of years} = 30$) underlying the test on linear regression coefficients, we decided to no longer report the p-values for the t-test formerly applied. Instead, we implemented a non-parametric test for the city-specific trends (Wilcoxon rank test), with greater statistical power due to the larger number of data points ($n = \text{number of cities} \times (\text{number of counterfactual temperatures series} + \text{factual temperature series}) = 15 \times (9+1) = 150$). We used this test to investigate whether there is a significant difference between city-specific results for the setup of with/without climate change, in line with the attribution focus of our study. In addition, we keep reporting the sign of the linear regression coefficients (with 95% confidence intervals) as an indication of the tendency with regard to positive or negative temporal trends (Table 2).

- It is understandable to want to keep the climate variability in the counterfactual analyses, but I would like to have more data and graphics on the attribution of pre-industrial to current time.

R4.5: As a response to this comment, we have made an additional effort to better describe the construction of counterfactual temperature series in the method section (II. 238-263). In the revised analysis, we used 1850-1900 as a pre-industrial reference period, which is a usual choice, see e.g. reference 48. We included two new supplementary figures, showing (i) the global mean surface temperature (GMST) data used for the detrending of observed city-specific temperature series (Supplementary Fig. 10), and (ii) the scatterplots and regression lines between annual GMST anomalies and city-specific warm-season temperatures (Supplementary Fig. 11). Furthermore, we newly included the R code applied to derive counterfactual temperatures into the code documentation, which will be made available via github/zenodo upon publication of the manuscript.

Reviewer #5 (Remarks to the Author):

Reviewer #6 (Remarks to the Author):

Thank you for the opportunity to review this interesting article. The manuscript is well written, researches a relevant topic and uses a novel approach to provide a better understanding on how climate change and other social changes shaped temperature related mortality in the past 30 years in Germany.

Here I provide some comments to the authors:

- The title could be improved to better convey the message of the paper. For instance, the authors could consider something on the lines of "Improvements in life expectancy compensate for increasing risks in heat-related excess mortality attributable to climate change" ;

R6.1: Implementing a new method based on counterfactual best linear unbiased predictors (ll. 298-305) we are now able to explicitly account for improvements in life expectancy in the revised analysis. The new title now reads: "Improvements in life expectancy mask rising trends in heat-related excess mortality attributable to climate change"

- In the abstract, the authors could stress more the implications of their findings that currently are not evident. Similarly, the authors could improve discussion of the implications in the discussion section as well. For instance, the authors could enlarge the discussion on how their findings could inform studies that project future trends in temperature related mortality.

R6.2: We agree that in the previous version of the manuscript we did not discuss well enough the implications of our findings. In the revised manuscript, we have included a new sentence on implications into the abstract (ll. 17-19). We have also added a short section on how our results could inform future projections studies of temperature-related mortality into the discussion part, as suggested by the reviewer (ll. 192-197).

- The authors use data from KNMI for temperature. However, the authors could provide further details on why this dataset is chosen compared to other gridded datasets such as ERA-5 or E-OBS provided by the Copernicus Data Store that might provide better coverage. For instance, the KNMI data has a large number of missing data on temperature.

R6.3: In the original version of the study, we used temperature data from KNMI because we wanted to use data from city-specific monitoring stations. Yet, considering this reviewer's comment (thank you very much!) we realized the disadvantages of using partly unhomogenized station data and we reconsidered our choice. In the revised analysis, we now apply daily mean temperature data extracted from ERA-5 Land starting in 1950. This choice has allowed us to include data from two additional cities into the analysis (Dortmund and Duisburg, see revised Table 1), which we formerly excluded because of inhomogeneities and large data gaps in the KNMI temperature data. Since we had to re-run the analysis based on the new temperature data, we used the opportunity to include more recent mortality data as well, allowing for an extension of the study period by two years.

- The authors used a different approach compared with previous studies in climate attribution. Additional information, should be provided in the methods section and/or in the appendix on their data management procedures and on the advantages and disadvantages compared with previous

approaches. Currently, it is not very easy to understand the steps they took and how sensitive these are to potential alternative choices.

R6.4: We agree on the poor documentation of the methods used to derive counterfactual temperatures in the previous version of the manuscript. In the revised version, we explain the methods in much greater detail (ll. 238-263). In that section, we also compare our approach to previous non-probabilistic attribution studies of heat-related mortality (references 5 and 6), which used a very similar method to derive counterfactual temperature series. In a sensitivity analysis, we have now also tested the alternative approach of using 5-year moving averages of global mean surface temperature (GMST) anomalies rather than the monthly data smoothed with singular spectrum analysis as done in our default setup (ll. 331-333; Supplementary Tables 5 and 7).

Furthermore, we have included two new figures in the supplementary material, documenting the global mean surface temperature (GMST) data derived from HadCRUT5 (Supplementary Fig. 10), and the linear regressions between GMST and city-specific warm-season temperature (Supplementary Fig. 11). Last but not least, we have incorporated the corresponding code in the R code documentation, which had previously been missing (because it was originally coded in Python).

Reviewer #6 (Remarks on code availability):

A Readme file is not currently present and could be added by the authors. The code has relevant title but could benefit from additional comments in some parts to better guide use. Also, as the data is not provided (at least in the files I reviewed) the authors could provide additional guidance in the code on where to access this.

R6.5: The R code provided so far has been anonymized for the purpose of the double-blind peer review. We plan to make the full code available via GitHub and zenodo upon publication. At that stage, we will also include a Readme file with specifications on where to access the raw data required to run the code. For now, we have revised and expanded the provided R code and we have added a few more comments as additional guidance for users.